# Use of Citizen Science-Derived Data for Spatial and Temporal Modeling of Particulate Matter near the US/Mexico Border

**Graeme N. Carvlin [1], Humberto Lugo [2], Luis Olmedo [2], Ester Bejarano [2], Alexa Wilkie [3], Dan Meltzer [3], Michelle Wong [3], Galatea King [3], Amanda Northcross [4], Michael Jerrett [5], Paul B. English [3,*], Jeff Shirai [1], Michael Yost [1], Timothy Larson [1,6] and Edmund Seto [1]** 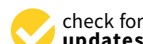

[1]  Department of Environmental and Occupational Health Sciences, University of Washington, Seattle, WA 98195, USA
[2]  Comite Civico del Valle, Brawley, CA 92227, USA
[3]  Tracking California, Public Health Institute, Richmond, CA 94804, USA
[4]  Department of Environmental and Occupational Health, George Washington University, Washington, DC 98824, USA
[5]  Department of Environmental Health Sciences, University of California, Los Angeles, CA 90095, USA
[6]  Department of Civil and Environmental Engineering, University of Washington, Seattle, WA 98195, USA
*  Correspondence: paul.english@cdph.ca.gov; Tel.: +1-(510)-620-3684

**Abstract:** This paper describes the use of citizen science-derived data for the creation of a land-use regression (LUR) model for particulate matter ($PM_{2.5}$ and $PM_{coarse}$) for a vulnerable community in Imperial County, California (CA), near the United States (US)/Mexico border. Data from the Imperial County Community Air Monitoring Network community monitors were calibrated and added to a LUR, along with meteorology and land use. $PM_{2.5}$ and $PM_{coarse}$ were predicted across the county at the monthly timescale. Model types were compared by cross-validated (CV) $R^2$ and root-mean-square error (RMSE). The Bayesian additive regression trees model (BART) performed the best for both $PM_{2.5}$ (CV $R^2 = 0.47$, RMSE = 1.5 µg/m$^3$) and $PM_{coarse}$ (CV $R^2 = 0.65$, RMSE = 8.07 µg/m$^3$). Model predictions were also compared to measurements from the regulatory monitors. RMSE for the monthly models was 3.6 µg/m$^3$ for $PM_{2.5}$ and 17.7 µg/m$^3$ for $PM_{coarse}$. Variable importance measures pointed to seasonality and length of roads as drivers of $PM_{2.5}$, and seasonality, type of farmland, and length of roads as drivers of $PM_{coarse}$. Predicted $PM_{2.5}$ was elevated near the US/Mexico border and predicted $PM_{coarse}$ was elevated in the center of Imperial Valley. Both sizes of PM were high near the western edge of the Salton Sea. This analysis provides some of the initial evidence for the utility of citizen science-derived pollution measurements to develop spatial and temporal models which can make estimates of pollution levels throughout vulnerable communities.

**Keywords:** $PM_{2.5}$; $PM_{coarse}$; land-use regression; community-based participatory research; citizen science; air sensors; community air monitoring

## 1. Introduction

Both $PM_{2.5}$ (particulate matter less than 2.5 µm in diameter) and $PM_{10}$ (particulate matter less than 10 µm in diameter) are linked to adverse health outcomes such as respiratory and cardiac disease, asthma, and increased mortality [1–4]. $PM_{coarse}$ is the difference between $PM_{10}$ and $PM_{2.5}$ and, as such, represents a different size fraction than $PM_{2.5}$, whereas $PM_{10}$ overlaps with $PM_{2.5}$. The United States (US) Environmental Protection Agency (EPA) refers to particles between 2.5 and 10 µm in diameter

as "PM$_{coarse}$" [5]. Studying PM$_{coarse}$ rather than PM$_{10}$ is, therefore, useful for understanding sources specifically associated with larger particle sizes.

PM differs spatially based on proximity to sources and air pollution transport. However, the ability to collect highly spatially resolved PM measurements is often limited by the number of monitors. This led to the use of modeling to estimate PM concentrations at locations and, for temporally varying data, at times when measurements are not available. One technique used to estimate PM exposure for a population is called land-use regression (LUR). LUR involves modeling PM using a combination of land-use variables such as traffic, area of land types, and distance to known sources. LUR models produce a continuous surface across the study area and can provide accurate PM exposure estimates [6,7]. Eeftens et al. used LUR modeling to predict PM$_{2.5}$ and PM$_{coarse}$ across 20 European cities, using the resulting predicted pollution surfaces to estimate PM exposure for a cohort involved in the European Study of Cohorts of Air Pollution Effects (ESCAPE) study [6]. Hoek et al. [7] reviewed 25 studies that used LUR models to predict air pollution levels including PM. Most studies employed a series of two-week sampling periods and estimated annual averages. The authors suggest that continuous sampling could help inform shorter-timescale models.

In this paper, a land-use regression model was developed for Imperial County, California (CA), a rural agricultural community in southeast California with a population of around 180,000 [8]. Imperial County has the second highest rate of childhood asthma emergency room visits in the state [9]. It is the home of a primarily Latino population (84%) and has some of the highest rates of unemployment (47%) and poverty (24%) in the nation [8]. The primary sources of air pollution are wind-borne dust, agricultural burning, mobile emissions from cars, trucks, and heavy-duty vehicles, and air pollution transport from Mexicali, Mexico, a city adjacent to the US/Mexico border [10–12]. The land-use regression model uses data from the Imperial County Community Air Monitoring Network ("the network"). The network was created through a collaboration between a local community group, Comite Civico del Valle (CCV), the Tracking California Program of the Public Health Institute, and academics at the University of California, Los Angeles (UCLA) and the University of Washington [13]. The network consists of 40 custom air quality monitors with Dylos particle counters (Dylos Corporation, Riverside, CA), which measure both PM$_{2.5}$ and PM$_{10}$, and a relative humidity and temperature sensor. The network is one of the largest community-based air monitoring networks in the US and is considered the first community-designed network of its size in the world [14]. Carvlin et al. showed an $R^2$ between the community monitor and regulatory monitors of 0.79 (hourly) and 0.84 (daily) for PM$_{2.5}$ and 0.78 (hourly) and 0.81 (daily) for PM$_{10}$ [15].

The LUR model described herein uses a combination of land-use variables and meteorology. Leave-one-out cross-validated root-mean-square error (RMSE) was used to compare between four different model types. After the model was selected, variable importance statistics were calculated. The goal of this LUR model was to create a two-dimensional (2D) air pollution map at the monthly scale for both PM$_{2.5}$ and PM$_{coarse}$. The predictions were compared to PM measurements made by regulatory monitors as an independent test of the LUR model.

## 2. Materials and Methods

### 2.1. Model Building

The particle counter used in the community monitors was a modified Dylos 1700. The firmware was changed to increase the number of particle size bins from two to four (>0.5 μm, >1.0 μm, >2.5 μm, and >10 μm). A custom circuit board was designed to interface the Dylos with an Arduino Yun to add networking capabilities. The circuit board also integrated a Honeywell HIH 6300 temperature and relative humidity sensor (Honeywell, Charlotte, NC, USA). Each Arduino Yun was then connected to either Wi-Fi, Ethernet, or a T-Mobile cellular modem.

In previous work, we collocated a monitor at the California Air Resources Board (CARB) Calexico-Ethel site, located at the Calexico High School on East Belcher Street, which collects reference

measurements of PM from both Federal Equivalent Method (FEM) beta-attenuation monitors (BAMs) and Federal Reference Method (FRM) filter-based gravimetric samplers. By comparing FEM and FRM data with data from the community air monitors, an equation for estimating mass concentrations from the Dylos particle count concentrations was developed. The calibration equation was validated by comparing calibrated monitor results with $PM_{2.5}$ measured by collocated reference instruments at six other sites. The process was previously described in detail [15].

Community monitoring $PM_{2.5}$, $PM_{10}$, relative humidity, and temperature data from 35 sites for a 12-month period, 1 October 2016 to 1 October 2017, were used in the following analyses. Relative humidity is known to change particle size due to the addition or subtraction of water from particles [16]. Since temperature and relative humidity were moderately correlated, only relative humidity was included in the conversion equation. The PM data were converted from particle counts to particle mass as detailed in Carvlin et al. [15]. However, the conversion equation was updated using data from the Calexico-Ethel site for the 12-month study period. The previous $PM_{10}$ conversion equation used data from 15 January 2016 to 12 July 2016. The conversion equation differed from the previous equation as we used new conversion constants based on the new time period: for $PM_{10}$, c1 = 9.35, c2 = 0.216, c3 = −0.344; for $PM_{2.5}$, c1 = 5.41, c2 = 0.00831, c3 = −0.224. $PM_{coarse}$ was calculated as $PM_{10}$ − $PM_{2.5}$. The respective conversion equations used for $PM_{2.5}$ and $PM_{10}$ were as follows:

$$Dylos_{bin1} = \beta_0 + \beta_1 BAM_{PM2.5} + \beta_2 RH + e(), \tag{1}$$

$$Dylos_{bin3} = \beta_0 + \beta_1 BAM_{PM10} + \beta_2 RH + e(), \tag{2}$$

where $\beta_0$ is the intercept, RH is the relative humidity as measured by the RH sensor on our custom circuit board, and e is the residual error. The factors c1, c2, and c3 were used in the inversion of the model to estimate BAM equivalent PM concentrations from Dylos measurements, as described in Carvlin et al. [15].

As a part of the conversion process, data were run through an automated quality control (QC) process; hours with less than 75% of data and data with particle counts less than 30 in Dylos bin 1 were discarded. After the automatic QC, a manual QC process was performed to identify time periods when the monitor response was slowly attenuated due to incremental dust build-up on the photodiode and when the monitor readings were oscillating rapidly between high and low, which resulted in a further 1.2% of data being dropped. The conversion equation can produce negative numbers, which were used as is in the following analyses unless otherwise noted. Hourly $PM_{2.5}$ and $PM_{coarse}$ data were averaged to monthly data using a 50% data completeness cutoff. This left a total of 207 monthly data points across 33 monitors. Each monitor had six months of data on average; however, some monitors had only a few months. The monitors that had the least amount of data were those near the Salton Sea. These monitors have poor cell reception and are subject to harsher environmental conditions, which leads to lower data completeness.

Regulatory data were downloaded from the California Air Resources Board (CARB) website. The regulatory network consists of five sites located near population centers in Imperial Valley that have Met One 1020 $PM_{10}$ beta attenuation monitors (BAMs) (Met One, Grants Pass, OR, USA) [17]. Two of these sites also have Met One 1020 $PM_{2.5}$ BAMs. There are also five sites located around the Salton Sea that are operated by the Imperial Irrigation District that have $PM_{2.5}$ and $PM_{10}$ Thermo Fischer Scientific Series 1405-D tapered element oscillating microbalances (TEOMs) (Thermo Fischer Scientific, Waltham, MA, USA) [18]. These sites were set up to monitor emissions from the Salton Sea as it recedes due to changes in water rights that reduced the agricultural runoff that kept the sea from evaporating. Only QC screened data were used. Values greater than 985 µg/m$^3$ for BAMs were excluded since this is above the range of the instrument [19]. No upper cutoff was used for TEOM measurements since all values were within the instrument range [20]. All negative values from regulatory instruments were kept as is and were included in the analysis.

Land-use variables and community monitor locations were loaded into ArcGIS (ESRI, v. 10.3.1, Redlands, CA, USA). Then, 250 m, 500 m, and 1000 m buffers were created around each monitor. Land-use parameters were sampled within each of these buffers. Geographic information system (GIS) and meteorological variables are listed in Table 1 along with their source, date, buffers, and averaging period. All data manipulation and analyses were performed using *R* statistical software (v. 3.3.3, https://www.r-project.org/).

**Table 1.** Model variables. RH—relative humidity; PM$_{2.5}$—particulate matter less than 2.5 μm in diameter; US—United states; CARB—California Air Resources Board.

| Data | Source | Date | Buffers (m) | Averaging Period |
|---|---|---|---|---|
| Latitude, longitude | Calculated | | | |
| Distance to the border, Salton Sea | Calculated | | | |
| Urban, types of farmland, types of crops, native and riparian vegetation (area) | Agricultural Land Use Survey (California Department of Water Resources), Farmland Mapping and Monitoring Program (California Department of Conservation) | 1997, 2012 | 250, 500, 1000 | |
| Small roads, large roads, railroads (length) | Topologically Integrated Geographic Encoding and Referencing (TIGER)/Line (US Census Bureau) | 2013 | 250, 500, 1000 | |
| Traffic | California Department of Public Health | 2013 | 250, 500, 1000 | Annual average of traffic counts |
| Agricultural burning (acres) | Imperial Air Pollution Control District | Study period | Sum within 5 km | Daily |
| RH, temperature | Community monitors | Study period | | Hourly |
| Wind direction, wind speed | Nearest government monitor (CARB) | Study period | | Hourly |
| Planetary boundary layer height | Rapid Refresh numerical weather model (RAP 130) (NOAA) | Study period | | Hourly |
| Satellite PM$_{2.5}$ | Randall Martin, Dalhousie University | 2002–2004 | | Every 3 years |

Agricultural burning records were received from the Imperial Air Pollution Control District. Acres burned was recorded on the daily level. This information was added to the model as acres burned within 5 km of a monitoring site within the last day. When multiple burns were recorded within 5 km of a site, they were summed.

Other GIS variables were considered but rejected since all monitors had the same value or nearly the same value for that variable. Dropped variables included indicators of industrial PM emissions since none of our monitors were located near industrial sites that had permits to release PM. Satellite PM$_{2.5}$ was included, but was not predictive, perhaps because the data were 15 years old and satellite measurements are known to not perform well in desert areas.

Meteorological data completeness was less than ideal, especially for planetary boundary layer height. Because the models require a complete dataset, hours that did not have meteorological data were dropped. This resulted in a limited number of complete hours for October and November 2017 and, therefore, they are not included in the monthly and yearly PM maps.

Some monitors have buffers that cross the US/Mexico border. However, we had no land-use data for Mexico. If only the US side of the land use was used, then the true value of that land use would be underestimated. To adjust for this, the percentage area of the buffer within Mexico was recorded for each site. Then, the variables were multiplied by $100/(100 - \%$ area), which gave the land use for the whole buffer assuming the same distribution in Mexico as in the US. A satellite image of the area showed that, in most cases, land within the buffer on both sides of the border was primarily urban land.

Land-use variables were converted from continuous to categorical or binary. This was done because the monitors do not cover the range of land use seen throughout the valley, particularly the range of land use sampled by the grid of points used for prediction (the fishnet). Therefore, if linear extrapolation was used then the fishnet predictions failed, becoming extremely small or large.

Histograms of each variable were analyzed to determine whether the variable should be converted to binary or categorical. The cut point for the binary variables was the first quartile. Most of these variables had nearly all of the measurements around zero and just a few at much higher values. All categorical variables were given three categories. The cut points for the categorical variables were the first quartile, the median, and the third quartile. If the continuous value was less than or equal to the first 25% of all values, it was given the categorical value "low"; if it was between 25% and 75%, it was categorized as "medium", and if it was greater than 75% it was categorized as "high". After conversion from continuous to categorical and binary, the fishnet predictions were much closer to the range of the monitoring measurements. However, the choice of cut points is dependent on the data and, therefore, limits the more general application of the models developed in this paper.

Three alternative models were considered. Categorical variables were converted to binary variables for use in models which cannot process categories. The models were a Bayesian additive regression trees (BART) model, a lasso model, and a partial least squares (PLS) model. PLS is a modeling technique that re-projects the data in order to find the dimension in the input variable space that explains the most variance in the outcome. PLS is used in PM modeling, in particular when there are a large number of variables [21]. Lasso is a penalized least squares method that reduces the number of variables in the model based on an alpha parameter. This parameter is chosen based on cross-validated testing. Mercer et al. [22] and Knibbs et al. [23] used lasso to help reduce the number of variables in PM LUR models. BART is a model that sums individual regression trees using a Bayesian approach [24]. It was used to predict torrential rain and avalanches, and to relate vehicle trip duration to household characteristics [25–27]. It should be noted that BART has a random component such that, each time it is re-run, it will produce slightly different results. This is why, for model creation and variable selection, it was important to have a large sample size of runs to get a sense of the average response for the model.

Models were compared by leaving out one site at a time and calculating the RMSE at that site using the rest of the sites. For each model, the RMSE was averaged across all sites. BART was found to be the best performing model for $PM_{2.5}$ and $PM_{coarse}$. A variable selection test for these models was performed to identify which variables had the most impact on the model. The variable selection for the BART models was done by dropping one variable at a time from the model and calculating the test $R^2$. Each BART model was run many times and the results were averaged to reduce bias from the random component. The variables that led to the largest decrease in $R^2$ were those that had the most impact on the model. The 10 most important variables for the $PM_{2.5}$ and $PM_{coarse}$ models were compared. In order to compare variable selection stability across models, the top 10 BART and lasso variables were compared. Lasso variables were selected based on their standardized coefficient values, corrected by their standard deviations.

*2.2. Prediction*

The same steps described in Section 2.1 were followed for a grid of points, the fishnet, equally spaced one mile (1.6 km) apart across all of Imperial County (i.e., sampling GIS parameters within buffers at each grid point, finding the nearest meteorological data, etc.).

The BART models were used to predict monthly $PM_{2.5}$ and $PM_{coarse}$ concentrations on the fishnet points. The residuals of the $PM_{2.5}$ and $PM_{coarse}$ BART models were kriged and added to the model predictions before mapping. This helped the model account for purely spatial variability. These combined PM levels, at both the monthly and study-average timescale, were kriged and a 2D surface was created and compared to kriged PM measurements.

The fishnet predictions were then compared to regulatory PM concentrations. The fishnet point closest to the regulatory site was chosen for comparison. The longest distance between a fishnet point and a regulatory site was <1 km. Model predictions and regulatory measurements were compared using $R^2$ and RMSE on the monthly and yearly timescales.

## 3. Results

Table 2 presents summary statistics for the community monitors averaged over the 40 sites in the network compared to the regulatory monitors. Regulatory sites are located to provide coverage of populated areas in the Imperial Valley, as well as to monitor the air quality around the Salton Sea, and they consist of a mix of BAM and TEOM instruments [14]. High hourly $PM_{2.5}$ and very high hourly $PM_{10}$ and $PM_{coarse}$ values were seen by both the community monitors and the regulatory monitors. The community monitors produced a similar county average to the regulatory monitors for $PM_{2.5}$. However, the community monitors measured slightly lower $PM_{10}$ and $PM_{coarse}$.

**Table 2.** Descriptive statistics for hourly $PM_{2.5}$, $PM_{10}$, and $PM_{coarse}$ (difference between $PM_{10}$ and $PM_{2.5}$).

| Statistic [1] | $PM_{2.5}$ (μg/m³) | | $PM_{10}$ (μg/m³) | | $PM_{coarse}$ (μg/m³) | |
|---|---|---|---|---|---|---|
| | **Dylos** | **Government** | **Dylos** | **Government** | **Dylos** | **Government** |
| Minimum | −6.5 | −3.0 | −7.7 | 0.9 | −22.2 | −5 |
| Median | 6.4 | 6.8 | 23.9 | 29.4 | 17.1 | 21.0 |
| Mean | 7.9 | 7.8 | 32.4 | 37.5 | 24.3 | 28.2 |
| Maximum | 52.3 | 149.4 | 1523.1 | 739.0 | 1109.5 | 780.0 |
| Number of hours | 4922 | | 4922 | | 4906 | |

[1] Matched hourly values averaged across all sites during the study period.

Table 3 shows the result of the modeling process for monthly $PM_{2.5}$. RMSE and $R^2$ were averaged across all of the leave-one-site-out variants. The lasso model chose 47 variables and the PLS model chose 24 variables. In terms of RMSE, lasso and PLS performed similarly and BART performed the best. The average RMSE between model predictions and the test set ranged from 1.50 to 1.79 μg/m³. Model $R^2$ values did not follow the same order as model RMSE. PLS had the highest $R^2$ (0.54), followed by PLS and BART (0.54 and 0.47, respectively). RMSE was used to select the top performing model since a lower prediction error was deemed more important than linearity. Also, there were a variable number of months of data for each site, and low sample size can dramatically change $R^2$, especially when the range of the data is small. For example, some sites only had two months of data and others only had a 2-μg/m³ variation in monthly average $PM_{2.5}$ over the course of a year. BART was chosen as the top performing model based on RMSE.

**Table 3.** Model selection monthly $PM_{2.5}$. PLS—partial least squares; BART—Bayesian additive regression trees; RMSE—root-mean-square error.

| Model | $R^2$ | RMSE |
|---|---|---|
| PLS | 0.54 [1] | 1.79 |
| Lasso | 0.51 | 1.70 |
| BART | 0.47 | 1.50 |

[1] The PLS model was not able to run for every site. Sites that could not run were dropped; $n = 207$.

The results of the modeling process for monthly $PM_{coarse}$ are presented in Table 4. Lasso chose 40 variables and PLS chose seven variables. Lasso and PLS performed similarly and BART performed the best. The average RMSE for $PM_{coarse}$ models ranged from 9.14 to 12.30 μg/m³. Model $R^2$ values followed the order of model RMSE for $PM_{coarse}$. The $R^2$ value for the $PM_{coarse}$ BART model was 0.65.

**Table 4.** Model selection monthly $PM_{coarse}$.

| Model | $R^2$ | RMSE |
|-------|-------|------|
| Lasso | 0.55 | 11.75 |
| PLS | 0.59 [1] | 11.48 |
| BART | 0.65 | 8.07 |

[1] The PLS model was not able to run for every site. Sites that could not run were dropped; $n = 207$.

The difference in performance between models was found to be unstable while testing differing predictor variables, i.e., the top model would switch between PLS, lasso, and BART depending on which variables were included. This would suggest, with this dataset, that these model types are more or less comparable in terms of performance.

Table 5 shows the results from the variable importance test for the $PM_{2.5}$ and $PM_{coarse}$ models. The top variable for the $PM_{2.5}$ model was relative humidity, and the top variable for the $PM_{coarse}$ model was temperature. These variables varied monthly and likely accounted for seasonal patterns. It should be noted the relative humidity may have a confounding effect since it was used to correct the Dylos measurements and was used in the model. However, the relative humidity data used in the model were a monthly average. The $PM_{2.5}$ model places more emphasis on meteorological variables, such as wind direction, wind speed, and planetary boundary layer height, and location, while the $PM_{coarse}$ model places more emphasis on land-use variables for farmland, native vegetation, and roads. The California Department of Water Resources defines native vegetation as all desert land and non-riparian land near bodies of water.

**Table 5.** Variable importance.

| Rank | PM$_{2.5}$ | | PM$_{coarse}$ | |
|------|------|------|------|------|
| | Name | Percentage Difference in $R^2$ (%) [1] | Name | Percentage Difference in $R^2$ (%) |
| 1 | Relative humidity | 16.1 | Temperature | 14.5 |
| 2 | Wind speed | 5.7 | Area of native vegetation, bottom tertile, 250 m | 8.4 |
| 3 | Wind direction, east | 5.5 | Area of fallow farmland, bottom 50%, 1000 m | 6.8 |
| 4 | Wind direction, northeast | 4.4 | Length of large roads, top tertile, 250 m | 6.6 |
| 5 | Length of large roads, middle tertile, 1000 m | 3.8 | Area of good farmland, top tertile, 250 m | 6.3 |
| 6 | Area of urban land, top tertile, 500 m | 3.7 | Area of prime farmland, middle tertile, 1000 m | 6.2 |
| 7 | Area of grain/hay crops, bottom tertile, 500 m | 3.6 | Length of railroads, bottom 50%, 1000 m | 5.8 |
| 8 | Planetary boundary layer height | 3.4 | Area of field crops, bottom tertile, 500 m | 5.6 |
| 9 | Longitude | 3.3 | Length of small roads, top tertile, 250 m | 5.6 |
| 10 | Length of small roads, top tertile, 1000 m | 3.3 | Length of large roads, middle tertile, 1000 m | 5.5 |

[1] The difference in test $R^2$ for a model that drops the selected variable compared to the model with all variables.

For $PM_{2.5}$, the lasso model chose land-use variables that represented nearly all categories of land use, as well as a variable representing railroad length, whereas the BART model put more emphasis on meteorological variables. For $PM_{coarse}$, the lasso model and BART models chose similar variables with more emphasis on land use in the lasso model and more emphasis on transportation in the BART model. The $PM_{coarse}$ lasso model also did not have any meteorological variables.

Figure 1a,b show the kriged study-average residuals from the $PM_{2.5}$ and $PM_{coarse}$ models. The $PM_{2.5}$ model underpredicted PM (−2.0 to −1.4 μg/m$^3$) in the area west of the Salton Sea and overpredicted PM (0.3 to 1.3 μg/m$^3$) in the area east of the Salton Sea. The $PM_{coarse}$ model underpredicted PM (−19.4 to −12.6 μg/m$^3$) in the area west of the Salton Sea. The $PM_{coarse}$ model overpredicted PM (1.2 to 4.0 μg/m$^3$) in the area east of the Salton Sea and in the south of Imperial Valley. The underprediction to the west of the Salton Sea was likely due to that area having high PM and similar land-use features to other sites. Moran's I (a measure of spatial autocorrelation) was used to see if there was spatial correlation in the residuals by month. $PM_{2.5}$ had two months where $p < 0.05$, May and August 2017, and $PM_{coarse}$ had three months with $p < 0.05$, January, July, and August 2017.

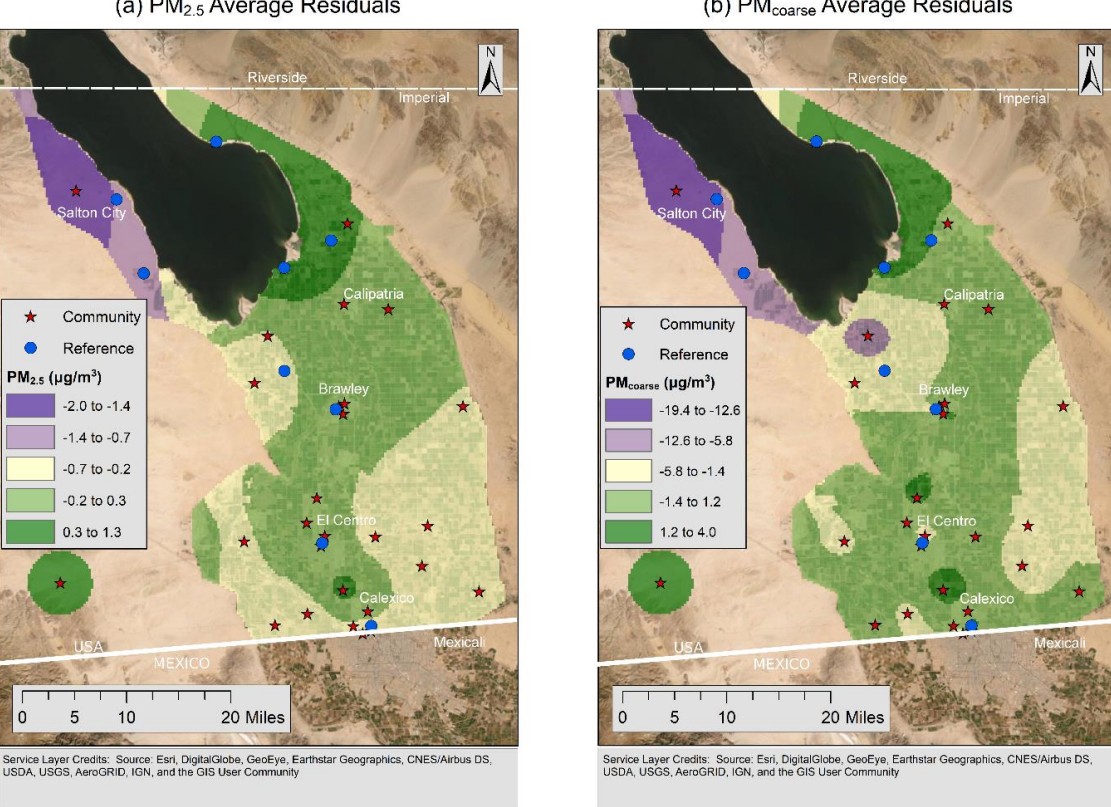

**Figure 1.** (**a**) PM$_{2.5}$ (particulate matter less than 2.5 μm in diameter) residuals; (**b**) PM$_{coarse}$ (difference between PM$_{10}$ and PM$_{2.5}$) residuals.

The average measured PM$_{2.5}$ over the study period is shown in Figure 2a and the average predicted PM$_{2.5}$ is shown in Figure 2b. The predicted PM$_{2.5}$ map is a combination of the PM$_{2.5}$ model and the kriged PM$_{2.5}$ residuals. The spatial structure of the predicted PM$_{2.5}$ map matches that of the measured PM$_{2.5}$ map. The highest average PM$_{2.5}$ levels were near the US/Mexico border and to the west of the Salton Sea near Salton City. Low-PM$_{2.5}$ regions appeared primarily to the east of the Salton Sea. The predicted PM$_{2.5}$ map is biased low compared to the measured PM$_{2.5}$ map.

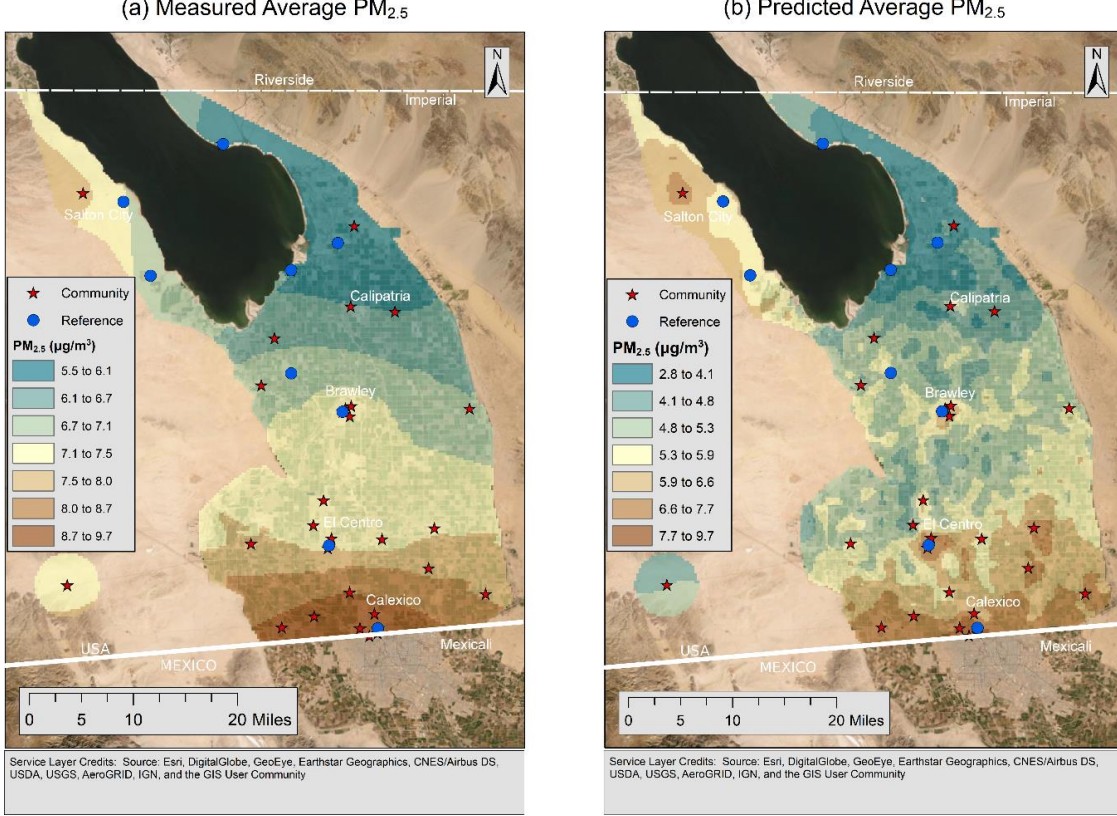

**Figure 2.** (**a**) Measured average PM$_{2.5}$; (**b**) predicted average PM$_{2.5}$.

Figure 3a shows the average measured PM$_{coarse}$ and Figure 3b shows the average predicted PM$_{coarse}$ over the study period. The predicted PM$_{coarse}$ map is a combination of the PM$_{coarse}$ model and the kriged PM$_{coarse}$ residuals. The spatial structure of the PM$_{coarse}$ map differs slightly from the measure PM$_{coarse}$ map in that the predictions are lower to the southeast of the Salton Sea and higher in the southeast portion of the Imperial Valley. The highest PM$_{coarse}$ to the west of the Salton Sea and the area east of the Salton Sea had low PM$_{coarse}$ concentrations. The predicted PM$_{coarse}$ map is biased low compared to the measured PM$_{coarse}$ map.

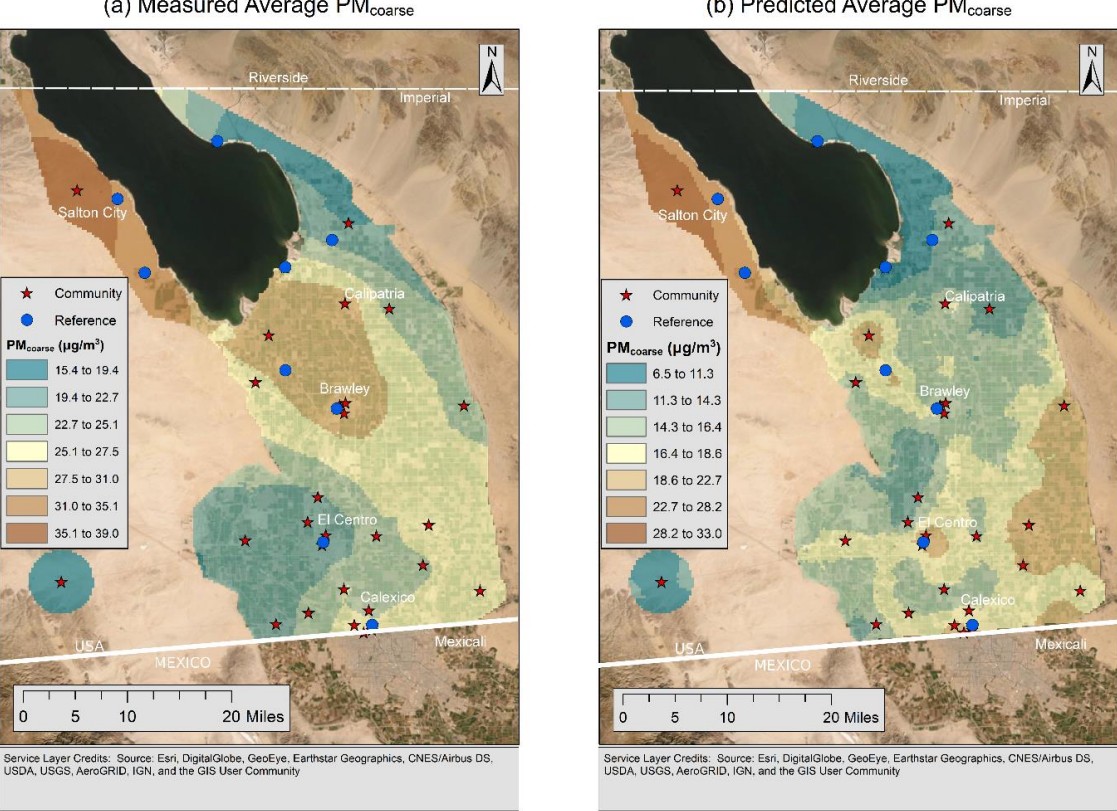

**Figure 3.** (**a**) Measured average PM$_{coarse}$; (**b**) predicted average PM$_{coarse}$.

Maps of measured and predicted monthly PM$_{2.5}$ are displayed in Figure 4a,b. Only monitors that had data for that month were used in each monthly map. There were not enough data to create maps for October and November 2016. The predicted PM$_{2.5}$ map is a sum of the PM$_{2.5}$ model and the kriged PM$_{2.5}$ residuals. The monthly PM$_{2.5}$ predictions are spatially similar to the monthly PM$_{2.5}$ measurements; however, the predictions are biased low. The highest PM$_{2.5}$ concentrations were seen in December 2016 and April through July 2017. PM$_{2.5}$ was high near the US/Mexico border throughout the year. PM$_{2.5}$ was high in the center of Imperial Valley near the city of Brawley in May 2017.

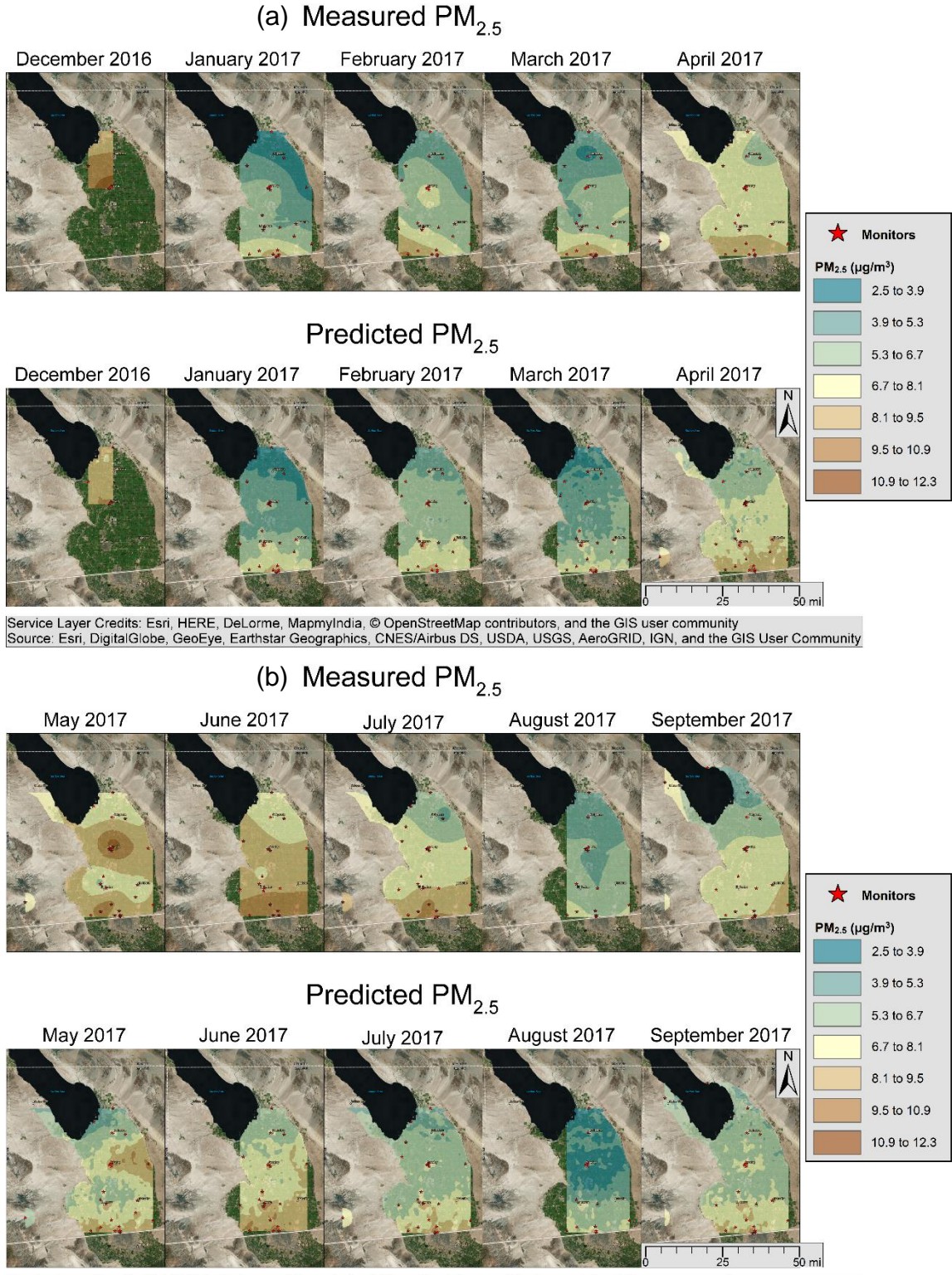

**Figure 4.** (**a**) Monthly measured and predicted PM$_{2.5}$ (December–April 2017); (**b**) monthly measured and predicted PM$_{2.5}$ (May–September 2017).

Figure 5a,b show maps of monthly measured and predicted PM$_{coarse}$. The predicted PM$_{coarse}$ map is a sum of the PM$_{coarse}$ model and the kriged PM$_{coarse}$ residuals. There was good spatial agreement between the PM$_{coarse}$ measurements and predictions; however, the predictions were biased

low. The highest PM$_{coarse}$ was seen around Brawley in May 2017 and in the southeastern part of Imperial Valley in September 2017.

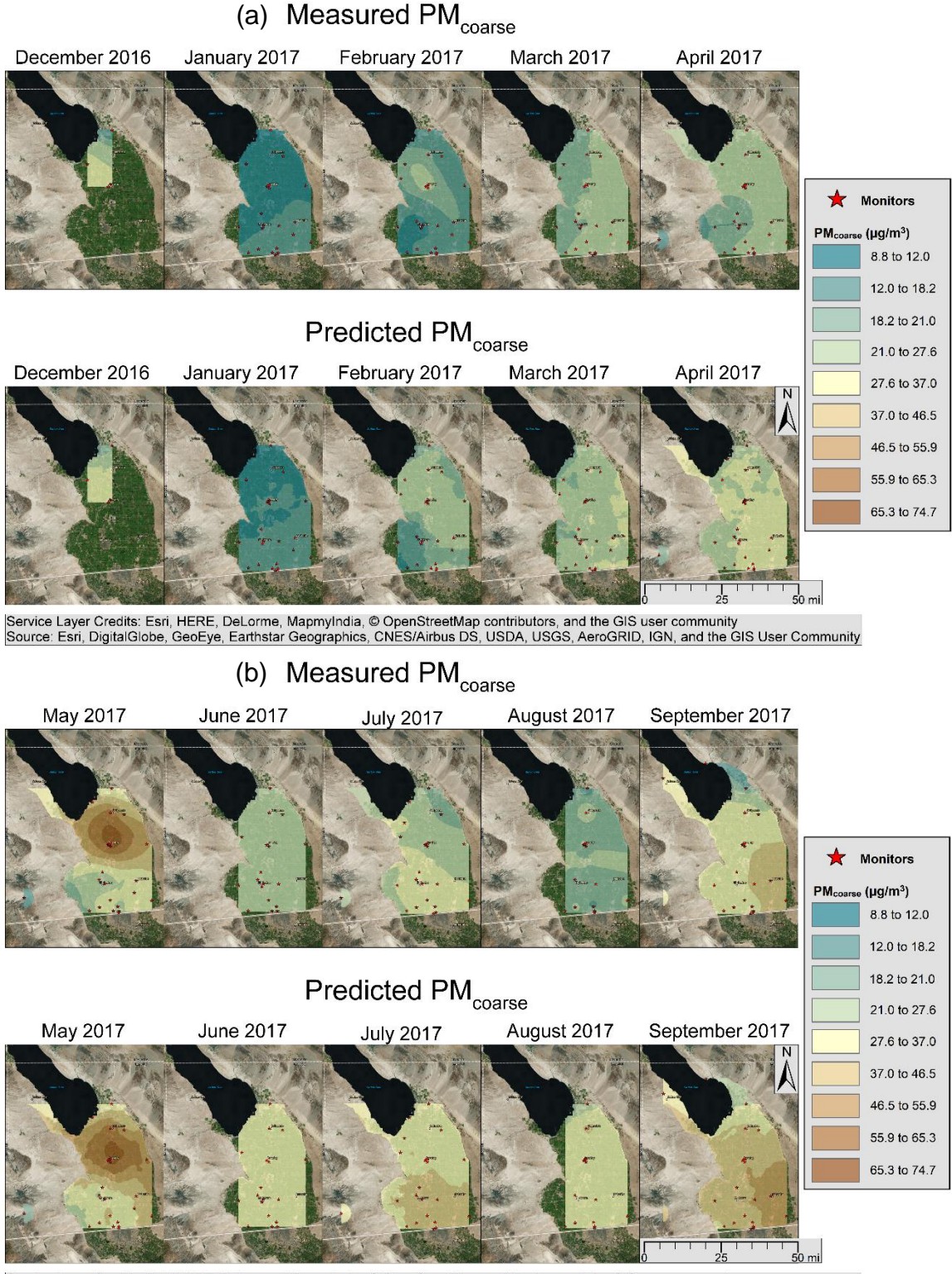

**Figure 5.** (**a**) Monthly measured and predicted PM$_{coarse}$ (December–April 2017); (**b**) monthly measured and predicted PM$_{coarse}$ (May–September 2017).

Table 6 shows summary statistics comparing the fishnet predictions and data from the regulatory monitoring network for monthly and yearly averages. The yearly RMSE was 3.2 µg/m$^3$ for PM$_{2.5}$ and 17.7 µg/m$^3$ for PM$_{coarse}$. The difference between the model prediction and regulatory monitor values is likely driven by spatiotemporal factors not accounted for in the model and measurement error introduced by the monitors and predictor variables. In particular, PM$_{coarse}$ measurements were biased low compared to regulatory measurements as noted above.

**Table 6.** Comparison of predictions with regulatory monitors, by monitor.

|  | PM$_{2.5}$ | | PM$_{coarse}$ | |
| --- | --- | --- | --- | --- |
|  | $R^2$ | RMSE | $R^2$ | RMSE |
| **Monthly** | 0.14 | 3.6 | 0.29 | 17.7 |
| **Yearly** | 0.51 | 3.2 | 0.36 | 16.3 |

## 4. Discussion

PM LUR models showed variable performance depending on model location and variable selection. The PM$_{2.5}$ LUR models developed in Eeftens et al. had validation $R^2$ values, comparing the model to validation data, of 0.21 to 0.78 with a mean of 0.59 [6]. Hoek et al. compared PM$_{2.5}$ LUR models from eight different studies and found model $R^2$ values, comparison of model performance without external validation data, of 0.17 to 0.73 with a mean of 0.52, and RMSE values of 1.1 to 2.3 µg/m$^3$ with a mean of 1.5 µg/m$^3$ [7]. All of these models produced annual averages of PM$_{2.5}$ and, therefore, are not directly comparable to the monthly models described in this paper. That said, the $R^2$ and RMSE values for the PM$_{2.5}$ models presented in this paper are within the range of the annual models found in the literature.

Eeftens et al. [6] created PM$_{coarse}$ LUR models for 20 European cities as a part of the ESCAPE project. The range of validation $R^2$ seen across the 20 models was 0.03 to 0.73, with a median $R^2$ of 0.57. Wolf et al. [28] used the ESCAPE data to create finer-spatial-resolution PM$_{coarse}$ LUR models for Germany, which had a validation $R^2$ of 0.49. Eeftens et al. [29] modeled PM$_{coarse}$ using an LUR model for eight areas in Switzerland. The average validation $R^2$ was 0.38. These studies were also annual averages. In general, PM$_{coarse}$ LUR models seem to have similar or slightly lower performance compared to PM$_{2.5}$ LUR models. The PM$_{coarse}$ models presented in this paper have $R^2$ values in the range of the annual models found in the literature.

Ahangar et al. [30] used 2017 data from the community network to obtain a finely resolved concentration map of PM using a dispersion model. The residuals between model estimates at the monitor locations and the measured concentrations were then interpolated to the grid points using kriging. They compared predicted monthly averaged PM with data from a regulatory monitor at one location and found that most of the modeled values fell within a factor of two of the regulatory values; their resulting concentration maps were consistent with this study in showing the highest values of PM at the international border, yet concentrations were higher directly south of the Salton Sea in their study as opposed to the west of the Salton Sea in the present study. These differences may be due to time period differences of the two studies and the different modeling approaches.

This paper helps to elucidate the effect of meteorological and land-use parameters on particulate matter in Imperial County. The variable selection process pointed to seasonality, as measured by relative humidity, meteorological parameters, and length of roads as prime drivers in monthly PM. Important variables for monthly PM$_{coarse}$ were seasonality, based on temperature, land-use variables for farmland, and length of roads. The predicted PM$_{coarse}$ map showed areas of high PM$_{coarse}$ around Brawley and to the west of the Salton Sea. The predicted PM map showed high levels in a large area near the US/Mexico border and to the west of the Salton Sea. This may point to agriculture as a potential source of PM$_{coarse}$, cross-border transport as a potential source of PM, and windblown dust as a source of both sizes of PM.

The PM and PM$_{coarse}$ model variable selection and predicted PM maps seem to agree with the emissions inventories in the 2018 PM$_{2.5}$ and 2018 PM$_{10}$ state implementation plans (SIPs) created by the Imperial County Air Pollution Control District (ICAPCD) [9,10]. The 2018 PM$_{2.5}$ SIP found that the main drivers of PM within Imperial County were unpaved road dust (39%), fugitive windblown dust (30%), off-road vehicles (8%), and agriculture (7%). This was seen in the models as a focus on seasonality and meteorology. The SIP argues that Imperial County would be in compliance with the National Ambient Air Quality Standards but for the transport of pollution from Mexico. As points of evidence, they cite emissions inventories for Calexico and Mexicali, Mexico that suggest that 60% of PM$_{2.5}$ in Calexico comes from Mexicali. Two other sites in the center of Imperial County, El Centro and Brawley, had an annual PM concentration of 7–8 µg/m$^3$ in 2016, whereas Calexico had an annual PM$_{2.5}$ concentration of 12.5 µg/m$^3$. This can be seen in the elevated measured and predicted PM$_{2.5}$ values at the US/Mexico border in Figure 2.

The 2018 PM$_{10}$ SIP found that the main drivers of PM$_{10}$ were fugitive windblown dust (75%), unpaved road dust (18%), and agriculture (3%). The PM$_{coarse}$ models, which can roughly be compared to the SIP's PM$_{10}$ emissions inventory, may have picked this up as seasonality and land use, specifically native vegetation, which was defined as primarily desert area. According to the SIP, wind patterns in Imperial County include high speed wind from the west, particularly during April and May, which may account for the high PM$_{coarse}$ levels seen during May 2017 in Figure 5.

In both Figures 2 and 3, high PM values can be seen west of the Salton Sea. The CARB review of the ICAPCD 2018 PM SIP remarks on this, saying that these high PM concentrations are due to dust from disturbed soils being swept into the air during high-wind events [31].

While there are challenges regarding calibration and accuracy of low-cost sensors, their low cost means they can be assembled into high-density networks that enable detailed modeling such as the LUR presented in this paper. Furthermore, the sensor and model data can be used help double-check and augment existing air monitoring and modeling efforts such as the emissions inventories in the Imperial County SIPs. If the necessary steps are taken to ensure data accuracy and proper display and explanation of sensor data, the authors see citizen science-derived data from low-cost air sensors as a valuable tool for communities, academic researchers, and government air-quality stakeholders.

**Author Contributions:** Conceptualization, G.N.C., A.N., M.J., and P.B.E.; data curation, H.L., E.B., D.M., and M.W.; formal analysis, G.C.; funding acquisition, P.B.E.; investigation, G.N.C., H.L., E.B., A.W., D.M., M.W., G.K., A.N., P.B.E., and J.S.; methodology, G.N.C., M.J., M.Y., T.L., and E.S.; project administration, A.W.; resources, E.B., and J.S.; supervision, M.Y., T.L., and E.S.; writing—original draft, G.N.C.; writing—review and editing, M.J., E.S., and P.B.E.

**Funding:** This research was funded by the National Institute of Environmental Health Sciences, under the NIH R01ES022722 grant. Jeff Shirai was partially supported by Award Number 5P30 ES007033-23 from the National Institute of Environmental Health Sciences.

**Acknowledgments:** The authors would like to acknowledge the work of the Community Steering Committee (CSC) and other study participants in assisting with site identification and selection, the air monitor hosts for permitting installation of the community air monitors on their properties, and the CSC in reviewing the IVAN website. We are also very appreciative of the help offered by the Technical Advisory Group (TAG)—a collection of government air-quality stakeholders and academics—who lent their expertise to the project. The California Air Resources Board (CARB) was instrumental to this work by offering their time, equipment, and vast knowledge of air pollution monitoring. We also thank Elena Austin, Kai Zhang, and Allan Hubbard for their suggestions for data analyses.

**Conflicts of Interest:** The authors declare no conflict of interest.

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
