# Peer review of "Use of Citizen Science-Derived Data for Spatial and Temporal Modeling of Particulate Matter near the US/Mexico Border"

_atmosphere, doi:10.3390/atmos10090495_

Round 1

Reviewer 1 Report

The manuscript is well-written and the argumentation and conclusions are reasonable. I suggest minor revisions in order to make some parts more clear and to improve readability for those not familiar with the studied area.

My major comments:

The community monitoring network presented in lines 80-98 and visualised in Figs. 1,2,3 is not sufficiently outlined, although the previous work of the authors, [13] is mentioned for details.  I was not able to indicate the monitoring sites, the names and the  Dylos main characteristics, without reading Ref.13. I suggest to expand the paragraph providing information on the calibration of the monitors,also to make the names in Figs.1-3 readable. Please, comment to which extent the conversion equation is different from the previous one ( derived on data from different time periods). As low cost sensors are usually very sensitive to humidity, some phrases explaining how this is treated in the data would be useful; Names of regulatory monitors are also mentioned, but it is difficult to figure out where they are located. Suggestion – add symbols for them in Figs. 1,2,3; The conversion of land use variables from continuous to categorical (lines 138-149) is not very clear. Providing Refs. where the reader could find more details, or giving a small example, would be beneficial. In Tables 2,3,4 provide the number of data used for the statistics (n) and precise if Tables 3,4 refer to hourly or monthly values.

Minor and technical remarks:

Lines 43, 61: transport -> air pollutant transport; Line 67: Dylos particle counters -> add (Producer) (Similarly to Met One in line 101); Line 75: 2D air pollution surface -> 2D air pollution map; Lines 76-77: regulatory models -> regulatory network or regulatory devices; Line 116 , TABLE 1: What is RAP 130 for planetary boundary layer height – provide Ref, or spell out; Lines 206-207: “low” sample size.. data is “small” Would it be possible to quantify this, even approximately; Lines 248-249: Not clear what is “Moran’s I” and “p”; Line 316: “PM LUR models” - check if this is not PM2.5 LUR models; Line 318: “ from the network” – precise community networ;  Line 323: highest elevations of PM – replace elevations with values; Line 325: to the east of the Salton Sea – it is perhaps to the west of the Salton Sea; Line 330: meteorology -> meteorological parameters; Lines 340 and 349 : there is inconsistency in the numbers, probably in line 340 it is not PM but PM2.5; Apply correct way for citing References; Use superscript for 3 in units for concentrations; Use uniform units (now both km and miles are used)

Reviewer 2 Report

Very interesting and well written manuscript. Final conclusions confirm the usefulness of low-cost sensors together with modeling methods as important tolls in air quality measurements conducted within national air monitoring. The only limitation in this type of works is the completeness of data and more specifically gaps in variables values having a potential impact on the concentration of pollutants at a particular site. Such comprehensive manuscripts are rarely found. Therefore I have only few less important comments

Abstract: RMSE - root-mean-square error please  expand the abbreviation

L39: Authors suggest that coarse PM are one between PM10 and PM2.5 while coarse mean generally all particles which originate from larger ones. I understand the methodological justification for choosing this fractions, however this should be pointed at the beginning of the manuscript

L41: „Studying PM coarse rather than PM10 is therefore useful for understanding sources specifically associated with larger particle sizes”.  Which sources do authors mean?
Many years of experience in the field of PM size fractionation allows me to conclude that many sources of PM emission – i.e. industrial,  construction and demolition activities,  road resuspension dust can produce both very coarse and PM10 particles. Therefore, any deduction regarding PM origin should not be based solely on grain size.

L57: with a population of around 175 0007 – provide citation

L65: UCLA - University of California, Los Angeles, please expand

L68 – please provide link to an official website which publishes data from this custom network

L80: data were collected from 35 sites? If yes please describe how mentioned 40 monitors are spread between those sites (one monitor one site?)

L84: please give the conversion equation and specify all variables and constants in this equation

L88: hours with less than 75% of data (authors mean 75% of data coverage?). What is the frequency (time resolution) of measurement by Dylos instrument?

L166: authors state that sensitivity analysis was performed by droppig one variable after another and testing a decrease in R2. Please enclose the initial variable database (for example in supplement)

L280: Fig 4 is unreadable. Please divide those maps into two separate figures – for example by season: non heating and heating one. Same regarding Fig 5.
